# Macrophages and Gut Barrier Function: Guardians of Gastrointestinal Health in Post-Inflammatory and Post-Infection Responses

**DOI:** 10.3390/ijms25179422

**Published:** 2024-08-30

**Authors:** Edward Xiangtai Meng, George Nicholas Verne, Qiqi Zhou

**Affiliations:** 1College of Medicine, University of Tennessee, Memphis, TN 38103, USA; 2Lt. Col. Luke Weathers, Jr. VA Medical Center, Memphis, TN 38105, USA

**Keywords:** macrophages, gut barrier, intestinal permeability, gastrointestinal health, post-inflammatory response, post-infection response, IBS

## Abstract

The gut barrier is essential for protection against pathogens and maintaining homeostasis. Macrophages are key players in the immune system, are indispensable for intestinal health, and contribute to immune defense and repair mechanisms. Understanding the multifaceted roles of macrophages can provide critical insights into maintaining and restoring gastrointestinal (GI) health. This review explores the essential role of macrophages in maintaining the gut barrier function and their contribution to post-inflammatory and post-infectious responses in the gut. Macrophages significantly contribute to gut barrier integrity through epithelial repair, immune modulation, and interactions with gut microbiota. They demonstrate active plasticity by switching phenotypes to resolve inflammation, facilitate tissue repair, and regulate microbial populations following an infection or inflammation. In addition, tissue-resident (M2) and infiltration (M1) macrophages convert to each other in gut problems such as IBS and IBD via major signaling pathways mediated by NF-κB, JAK/STAT, PI3K/AKT, MAPK, Toll-like receptors, and specific microRNAs such as miR-155, miR-29, miR-146a, and miR-199, which may be good targets for new therapeutic approaches. Future research should focus on elucidating the detailed molecular mechanisms and developing personalized therapeutic approaches to fully harness the potential of macrophages to maintain and restore intestinal permeability and gut health.

## 1. Introduction

The gut barrier is a complex and dynamic system essential for maintaining gastrointestinal (GI) health and overall homeostasis. It serves as a critical interface between the external environment and the body, orchestrating a delicate balance between defense against harmful pathogens and toxins, while facilitating the absorption of nutrients [1,2]. This barrier is composed of several key components, each of which contributes to its multifaceted protective role.

At the forefront, epithelial cells form a continuous resilient lining of the gut wall. These cells are tightly connected by specialized structures known as tight junctions, which regulate the paracellular transport of ions and small molecules. The integrity of these tight junctions is vital because they prevent the uncontrolled passage of potentially harmful substances, thereby maintaining the selective permeability of the gut barrier [3]. Any compromise in the tight junctions can lead to increased intestinal permeability, a condition often referred to as a “leaky gut”, which has been implicated in various GI disorders, including inflammatory bowel disease (IBD) and irritable bowel syndrome (IBS) [4].

The mucus layer, a critical component of the gut barrier that serves as the first line of defense against microbial invaders, overlies the epithelial layer. This viscous layer, produced by goblet cells, is rich in antimicrobial peptides such as defensins and immunoglobulins such as IgA. These molecules work synergistically to neutralize pathogens, preventing them from reaching the epithelial cells and penetrating deeper into tissues [5]. Moreover, the mucus layer supports a symbiotic relationship with the gut microbiota, a diverse community of microorganisms that plays a crucial role in digestion, nutrient production, and immune modulation. The microbiota also competes with pathogens for resources and space, further enhancing the defensive capabilities [6]. Disruptions in the gut barrier through physical damage, microbial imbalance, or immune dysfunction can lead to a cascade of adverse effects, including chronic inflammation, infection, and the development of autoimmune conditions [7].

## 2. Research Gap and Objective

Although the structural and functional components of the gut barrier are well characterized, there is a significant gap in our understanding of the specific roles that immune cells, particularly macrophages, play in maintaining and restoring this barrier during health and disease. Macrophages, which originate from monocytes in the bloodstream, are a vital part of the innate immune system and strategically positioned throughout the gut mucosa. These cells are highly versatile and exhibit remarkable plasticity, allowing them to adapt their functions to various environmental cues [8].

Macrophages perform several critical functions in gut health. They are key players in the clearance of apoptotic cells and cellular debris, which prevents unnecessary inflammation and promotes tissue homeostasis. Macrophages are also involved in antigen presentation, where they process and present antigens to T cells, thereby initiating and regulating adaptive immune responses [9,10]. This function is particularly important in the gut, where the immune system must strike a balance between responding to harmful pathogens and tolerating benign dietary antigens and commensal microbes. Additionally, macrophages secrete a range of cytokines that orchestrate inflammatory responses and facilitate tissue repair. These cytokines can either promote inflammation, as observed with pro-inflammatory cytokines such as TNF-α and IL-6, or suppress it through anti-inflammatory cytokines such as IL-10 and TGF-β, depending on the specific needs of the gut environment [11].

Despite these well-documented functions, the precise mechanisms by which macrophages contribute to gut barrier integrity, particularly during periods of inflammation and infection, remain unclear. For instance, it is not fully understood how macrophages transition between their pro-inflammatory (M1) and anti-inflammatory (M2) phenotypes in response to different stimuli and how these transitions impact the function of the gut barrier. Moreover, the interactions between macrophages and other cells within the gut, including epithelial cells and microbiota, are complex and likely play a crucial role in determining the outcome of inflammatory and infectious events [12].

This review seeks to bridge these knowledge gaps by providing a comprehensive analysis of the multifaceted roles of macrophages in maintaining the gut barrier function. We will explore their involvement in post-inflammatory and post-infectious responses, emphasizing their critical importance in preserving GI health. Additionally, this review highlights the potential of macrophage-targeted therapies for treating GI disorders while also considering the challenges and risks associated with such approaches. This review aims to contribute to a deeper understanding of the immune regulation of the gut barrier and identify new avenues for therapeutic intervention.

## 3. Role of Macrophages in Maintaining Gut Barrier Function

### 3.1. Types of Macrophages

**Resident Macrophages:** These macrophages are permanently embedded in the gut mucosa and play a fundamental role in maintaining gut homeostasis and epithelial barrier integrity. These cells are strategically located within the mucosal layer, allowing them to perform several key functions that are essential for gut health.

One of their primary functions is the clearance of apoptotic cells and cellular debris. By efficiently removing these dead cells, resident macrophages prevent the accumulation of potentially inflammatory materials and maintain tissue health [13]. This phagocytic activity is crucial for preventing unnecessary inflammation and ensuring a clean and functional gut lining.

Resident macrophages are also significant producers of anti-inflammatory cytokines such as interleukin-10 (IL-10). IL-10 helps modulate immune responses by suppressing excessive inflammatory activity and promoting a tolerant environment within the gut [14]. This cytokine contributes to the regulation of local immune responses, thereby preserving the integrity of the epithelial barrier and preventing damage caused by chronic inflammation.

Resident macrophages play a vital role in maintaining tight junction stability between epithelial cells. They secrete various factors that support the formation and maintenance of tight junctions, which are essential for preserving the impermeability of the gut barrier and preventing the translocation of harmful substances and pathogens [15,16,17].

In addition to their direct interactions with epithelial cells, resident macrophages are integral to maintaining a balanced gut microbiota. They interact with diverse communities of microorganisms residing in the gut, contributing to microbial homeostasis and influencing the composition of the gut microbiota. This interaction is crucial for sustaining overall gut health and preventing dysbiosis, which can lead to various gastrointestinal disorders [18,19,20,21].

Resident macrophages are essential for preserving a stable and tolerant gut environment. Their multifaceted roles in tissue homeostasis, the modulation of inflammation, and microbial balance underscore their importance in long-term gut health.

**Infiltrating Macrophages:** Infiltrating macrophages are recruited from the bloodstream to sites of inflammation or infections. Unlike resident macrophages, these cells are not permanently located in the gut but are mobilized in response to acute challenges, including tissue damage and microbial invasion [22,23].

Upon entering the affected tissue, infiltrating macrophages adapt their phenotype to the local environment. Initially, they often adopt a pro-inflammatory (M1) phenotype. In this state, they produce a range of cytokines including TNF-α, IL-6, and IL-1β, which are crucial for initiating and amplifying the inflammatory response. These cytokines help recruit additional immune cells to the site of infection or injury and enhance the macrophages’ phagocytic activity to eliminate pathogens and damaged cells [10,24,25].

However, to prevent excessive tissue damage and chronic inflammation, infiltrating macrophages can transition to an anti-inflammatory (M2) phenotype. M2 macrophages secrete cytokines, such as IL-10 and TGF-β, which play a key role in resolving inflammation, promoting tissue repair, and restoring epithelial barrier integrity [26]. This phenotypic plasticity is essential for balancing the immune response and ensuring the effective resolution of inflammation.

Infiltrating macrophages also interact with other immune cells and the gut microbiota to modulate the overall immune response. They help coordinate the response to pathogens and contribute to the re-establishment of microbial balance following an inflammatory event. Their ability to dynamically adjust their functions in response to environmental cues allows them to effectively manage both acute and resolving inflammatory processes [27,28,29,30].

### 3.2. Functions

#### 3.2.1. Barrier Maintenance

Macrophages play a pivotal role in maintaining the integrity and functionality of the gut barrier, which is essential for protecting the body against pathogens and maintaining overall GI health [31,32]. Their contributions to epithelial repair and regeneration are particularly significant [33,34]. One of the key ways macrophages support barrier maintenance is through the secretion of growth factors such as epidermal growth factor (EGF) and transforming growth factor-beta (TGF-β) [35]. These growth factors are crucial for stimulating epithelial cell proliferation and differentiation, which are vital for the repair and regeneration of the epithelial lining after injury or inflammation [36]. Additionally, cytokines such as IL-10 produced by macrophages help maintain tight junction integrity and intestinal permeability, which are crucial for barrier function [37].

EGF promotes the rapid proliferation of epithelial cells, facilitating wound closure and the restoration of the gut barrier [38,39,40]. TGF-β, on the other hand, not only aids in cell proliferation but also plays a role in the differentiation of epithelial cells, ensuring that newly formed cells are properly integrated into the existing tissue architecture. This dual action of promoting both proliferation and differentiation makes these growth factors indispensable for enhancing barrier resilience and maintaining gut integrity [41,42].

In addition to growth factors, macrophages produce anti-inflammatory cytokines such as interleukin-10 (IL-10). IL-10 is a critical regulator of immune responses and plays a significant role in maintaining tight junction integrity and intestinal permeability [43]. Tight junctions are protein complexes that seal the spaces between epithelial cells, preventing the passage of harmful substances, while allowing essential nutrients to pass through. IL-10 helps stabilize these tight junctions, reduce intestinal permeability, and protect the gut barrier from potential breaches.

By modulating immune responses and promoting a balanced environment, IL-10 also helps prevent chronic inflammation, which can compromise barrier function [14]. Chronic inflammation often leads to the breakdown of tight junctions, increasing intestinal permeability, and allowing pathogens and toxins to enter the bloodstream. By maintaining tight junction integrity, IL-10-producing macrophages help to avert such detrimental outcomes, ensuring that the gut barrier remains robust and functional [44,45].

Moreover, macrophages interact with other cell types in the gut, including fibroblasts and endothelial cells, to coordinate repair and regeneration processes [46]. They release a variety of signaling molecules that enhance the recruitment and activation of these cells, further supporting tissue repair and barrier maintenance. This collaborative effort ensures a comprehensive approach to maintaining gut health, addressing not only immediate damage, but also promoting long-term resilience.

Macrophages are indispensable for the maintenance of the barrier in the gut. Through the secretion of growth factors such as EGF and TGF-β, they drive epithelial repair and regeneration. Additionally, the production of IL-10 and other cytokines helps to maintain tight junction integrity and regulate intestinal permeability. These multifaceted roles underscore the critical importance of macrophages in preserving the gut barrier, highlighting their potential as therapeutic targets for enhancing gastrointestinal health and treating gastrointestinal disorders.

#### 3.2.2. Immune Modulation

Macrophages are central to immune regulation within the gut and play a crucial role in mitigating excessive immune activation [47]. By producing anti-inflammatory cytokines, such as interleukin-10 (IL-10) and transforming growth factor-beta (TGF-β), macrophages suppress inflammatory responses that could otherwise compromise gut barrier integrity. These cytokines help create an anti-inflammatory environment that protects the gut lining from damage and promotes tissue repair and regeneration [48].

In addition to their anti-inflammatory functions, macrophages also play a pivotal role in antigen presentation to T cells. This process is essential for orchestrating a balanced immune response that ensures effective pathogen clearance, while safeguarding tissue homeostasis [49]. Macrophages process and present antigens to T cells, facilitating their activation and differentiation [50,51]. This interaction is critical for mounting an appropriate immune response that targets pathogens without causing excessive inflammation, which could harm the gut epithelium [52].

By balancing pro-inflammatory and anti-inflammatory signals, macrophages help maintain gut homeostasis, protecting against chronic inflammation and its associated damage [32,53]. Their ability to modulate immune responses makes them key players in the maintenance of gut health, highlighting their potential as targets for therapeutic interventions aimed at treating GI disorders characterized by dysregulated immune activity.

## 4. Post-Inflammatory and Post-Infection Responses

### 4.1. Inflammation Resolution

#### 4.1.1. Macrophage Phenotype Switch

Macrophages exhibit remarkable plasticity, allowing them to transition from a pro-inflammatory (M1) to an anti-inflammatory (M2) phenotype. This ability is crucial for resolving inflammation and facilitating tissue healing [23,54]. During the initial response to infection or injury, macrophages adopt the M1 phenotype, which is characterized by the production of pro-inflammatory cytokines such as TNF-α, IL-6, and IL-1β. M1 macrophages are essential for pathogen elimination and the initiation of inflammatory cascades that help control infections and remove damaged cells.

M1 macrophages also secrete reactive oxygen species (ROS) and nitric oxide (NO), which are effective in killing pathogens and signaling other immune cells to the infection [55,56]. Their heightened phagocytic activity and pro-inflammatory nature are critical for immediate defense against invading microbes and for triggering the subsequent immune responses [57,58].

As the initial threat subsides, inflammatory responses must be tempered to prevent excessive tissue damage and chronic inflammation. Thus, the switch to the M2 phenotype is crucial. M2 macrophages are characterized by their anti-inflammatory and tissue repair functions. They secrete anti-inflammatory cytokines, including IL-10 and TGF-β, which help suppress ongoing inflammation and promote a healing environment [59].

In addition to cytokines, M2 macrophages produce growth factors such as vascular endothelial growth factor (VEGF) and fibroblast growth factor (FGF) [60]. These growth factors are instrumental in the promotion of angiogenesis and tissue regeneration. VEGF stimulates the formation of new blood vessels, enhancing blood supply and nutrient delivery to damaged tissues, whereas FGF supports the proliferation and differentiation of fibroblasts and epithelial cells, which are crucial processes for wound healing and the restoration of tissue integrity [61].

M2 macrophages also play a role in remodeling the extracellular matrix (ECM), which is a critical component of tissue repair. They produce enzymes such as matrix metalloproteinases (MMPs), which help degrade and remodel the ECM, facilitating the replacement of damaged tissue with new functional cells [62].

#### 4.1.2. Potential Side Effects of Modulating Macrophage Activity

While targeting macrophage plasticity to promote the M1-to-M2 transition presents a promising therapeutic approach, it is not without risks. The overactivation of the M2 phenotype may lead to excessive tissue remodeling, fibrosis, or an impaired immune response. For instance, an overly robust M2 response could contribute to fibrotic conditions by promoting excessive ECM deposition and scarring, which could compromise tissue function [63]. Additionally, a premature or excessive shift to the M2 phenotype might dampen the necessary immune response, allowing pathogens to persist or leading to the insufficient clearance of damaged cells [64,65].

Moreover, sustained M2 activity can suppress the immune system, potentially increasing the risk of secondary infections or the development of tumors due to reduced immune surveillance [64,65]. Therefore, while modulating macrophage activity offers therapeutic potential, it must be carefully balanced to avoid unintended consequences that could exacerbate or cause new health issues. Understanding the fine tuning required for macrophage modulation is essential for developing safe and effective treatments for conditions involving chronic inflammation and impaired healing.

### 4.2. Tissue Repair and Regeneration

#### Wound Healing

M2 macrophages play a crucial role in wound repair and tissue regeneration following inflammation or infection. They release growth factors, such as VEGF and FGF, which stimulate epithelial cell migration, proliferation, and differentiation [63]. This cellular activity is crucial for restoring the structural integrity of damaged tissues and for ensuring effective wound closure. The coordinated actions of these growth factors lead to the re-epithelialization of wounds, thereby forming a new protective barrier and preventing further infection or injury.

In addition to their role in epithelial repair, M2 macrophages significantly contribute to angiogenesis and the formation of new blood vessels, which are essential for sustaining the healing process [64,65]. By promoting angiogenesis, M2 macrophages enhance the delivery of vital nutrients and oxygen to regenerated tissue, thus accelerating the healing process and improving tissue quality. This newly formed vasculature not only supports the metabolic demands of the healing tissue but also facilitates the removal of waste products [66]. The dual role of M2 macrophages in epithelial repair and angiogenesis underscores their importance in wound healing and highlights their potential as therapeutic targets for improving wound healing outcomes.

## 5. Microbial Regulation

### Microbiota Restoration and Macrophage–Microbiome Interactions

Macrophages play a pivotal role in maintaining microbial balance within the gut, particularly following episodes of inflammation or infection [67,68]. Their functions include both direct and indirect interactions with the gut microbiome.

**Direct Interactions:** Macrophages interact directly with microbiota through various mechanisms. They engage in the phagocytosis of pathogens and cellular debris, which helps clear harmful microorganisms from the gut environment. This action is crucial for preventing the persistence of pathogenic microbes and for mitigating inflammation. Macrophages also produce antimicrobial peptides that target and eliminate pathogenic bacteria, contributing to the immediate defense against infections [69].

Additionally, macrophages can influence microbial communities through the secretion of cytokines and other signaling molecules [10]. Macrophages can release cytokines that affect the growth and activity of specific microbial populations. This interaction can either suppress pathogenic bacteria or promote the proliferation of beneficial microbes [26]. The balance between these cytokines is essential for maintaining a healthy microbial ecosystem and supporting gut health.

**Indirect Interactions:** Macrophages create an environment that supports the growth of beneficial microbes. They modulate local immune responses by secreting factors that promote the survival and colonization of commensal bacteria. By maintaining a stable and balanced microbial community, macrophages help preserve the gut barrier integrity and support immune homeostasis [70,71]. This role is particularly important in preventing pathogen translocation and ensuring the proper functioning of the gut.

Recent research has highlighted the complexity of macrophage–microbiome interactions, emphasizing the reciprocal influence between these two components. For instance, the gut microbiome can shape macrophage function and polarization. Commensal bacteria produce metabolites such as short-chain fatty acids (SCFAs) that influence macrophage behavior. SCFAs can modulate macrophage polarization towards an anti-inflammatory phenotype, thereby promoting a balanced immune response and tissue repair [72,73,74].

**Fungi and Microbiota-Based Therapies:** Emerging evidence also underscores the importance of including fungi in discussions about microbiota-based therapies. Fungi such as the medicinal mushroom *Hericium erinaceus* have shown potential in modulating gut inflammation and influencing macrophage activity. Studies suggest that *Hericium erinaceus* can downregulate pro-inflammatory cytokines, thereby affecting macrophage responses and contributing to reduced inflammation [75,76]. This highlights the potential of integrating fungal components into therapeutic strategies aimed at managing gut inflammation and restoring microbial balance.

Incorporating fungi into microbiota-based therapies is an exciting avenue for future research. By leveraging the therapeutic properties of fungi, such as *Hericium erinaceus*, researchers may develop novel strategies to address dysbiosis and associated gastrointestinal disorders. These approaches could complement existing bacteria-based therapies and offer additional tools for managing gut health.

Overall, macrophages are central to complex interactions between immune responses and microbial communities in the gut. Their ability to directly and indirectly influence microbial balance, coupled with emerging evidence about fungal contributions, underscores their critical role in maintaining gut health. Understanding these interactions provides valuable insights into potential therapeutic approaches that integrate bacterial and fungal components to restore and maintain a healthy microbiome.

## 6. Macrophages in Gastrointestinal Diseases

### Infections and Inflammatory Conditions


**Pathogen Clearance:**


Macrophages are integral to the immune response in GI infections, demonstrating remarkable adeptness in recognizing, engulfing, and eliminating pathogens through various mechanisms. One of the primary processes utilized is phagocytosis, in which macrophages engulf and digest microbial invaders. This is complemented by the production of reactive oxygen species (ROS) and nitric oxide (NO), which are potent antimicrobial agents that aid in the destruction of pathogens within the phagolysosomes of macrophages [77].

In addition to their direct pathogen-killing activities, macrophages play a crucial role in orchestrating a broader immune response. They secrete a range of pro-inflammatory cytokines, including tumor necrosis factor-alpha (TNF-α), interleukin-1 beta (IL-1β), and interleukin-6 (IL-6) [78]. These cytokines serve as signaling molecules that recruit and activate other immune cells such as neutrophils and lymphocytes, thereby amplifying the immune response against infections. By initiating and sustaining an inflammatory environment, macrophages ensure that the immune system remains engaged and effective in eradicating the invading pathogens. Their dual roles in both direct pathogen clearance and immune system modulation underscore the importance of macrophages in maintaining gastrointestinal health and combating infections.


**Irritable Bowel Syndrome (IBS):**


In conditions such as IBS, macrophages play a crucial role in modulating gut inflammation and visceral sensitivity [79]. Following an inflammatory or infectious event, the restoration of the gut barrier function is crucial for preventing chronic inflammation and further damage. Macrophages play a pivotal role in the recovery process by clearing apoptotic cells, resolving inflammation, and promoting tissue repair. Their plasticity allows them to adapt to the changing microenvironment, switching from a pro-inflammatory to an anti-inflammatory phenotype as needed. This ability is particularly important in IBS, where chronic inflammation and impaired barrier function are common.

IBS is a disorder characterized by abdominal pain, bloating, and altered bowel habits. The gut barrier often exhibits increased permeability, also known as the “leaky gut”. This compromised barrier can allow pathogens and antigens to penetrate the intestinal lining, triggering immune responses and contributing to the persistent inflammation observed in patients [79,80]. Through their dynamic ability to transition between different functional states, macrophages are essential for mitigating these effects.

When the gut barrier is damaged, pro-inflammatory macrophages (M1) are activated to combat pathogens and clear debris. However, the prolonged activation of M1 macrophages can exacerbate inflammation, leading to further barrier disruption [16]. Therefore, switching to an anti-inflammatory (M2) phenotype is critical. M2 macrophages secrete anti-inflammatory cytokines and growth factors, which facilitate tissue repair and strengthen the epithelial barrier. This transition not only helps resolve inflammation but also promotes the restoration of normal gut barrier function, thereby reducing symptoms and preventing relapse in patients [81,82].

Additionally, macrophages interact with the gut microbiota, which plays a significant role in IBS pathophysiology. Dysbiosis, or microbial imbalance, is often observed in patients with IBS and can influence macrophage behavior [83,84]. By recognizing and responding to microbial signals, macrophages help maintain a balanced microbiota, which is crucial for gut health [85]. Therapeutic strategies that enhance beneficial macrophage–microbiota interactions could potentially restore microbial equilibrium and improve gut barrier function in patients with IBS.

Understanding the specific mechanisms by which macrophages contribute to these processes in IBS can provide insight into potential therapeutic strategies. For instance, treatments aimed at modulating macrophage polarization to favor the anti-inflammatory M2 phenotype or enhancing their ability to interact with the microbiota could offer new avenues for managing IBS. As research continues to elucidate these complex interactions, targeted therapies that harness the protective and reparative functions of macrophages may emerge, offering hope for more effective IBS treatments and improved patient outcomes.

While IBS is primarily a functional GI disorder characterized by chronic abdominal pain and altered bowel habits, emerging evidence suggests a role for low-grade inflammation and immune activation, including macrophage involvement, in its pathogenesis [86]. Targeting therapies for macrophage activation and polarization could potentially alleviate the symptoms and improve gut function in patients with IBS.


**Inflammatory Bowel Disease (IBD):**


In IBD, including Crohn’s disease and ulcerative colitis, macrophages play a critical role in the inflammatory processes that characterize these chronic conditions [87]. Macrophages are involved in both the initiation and perpetuation of inflammation in IBD. They are activated in response to various stimuli, including microbial components and host-derived signals, leading to the production of pro-inflammatory cytokines and chemokines [88]. This results in the recruitment and activation of additional immune cells, contributing to the chronic inflammatory environment observed in IBD.

Macrophages in IBD often display an altered functional state, with an increased production of inflammatory mediators, such as TNF-α, IL-1β, and IL-6. This dysregulated macrophage response can lead to tissue damage and disease exacerbation [88]. Additionally, macrophages in IBD may exhibit an impaired resolution of inflammation, resulting in sustained inflammatory responses and persistent tissue damage [89].

Recent studies have highlighted the potential of targeting macrophage-mediated inflammation in IBD. For example, therapies aimed at modulating macrophage activation or promoting the resolution of inflammation could offer new treatment options for IBD patients. Understanding the specific mechanisms by which macrophages contribute to IBD pathogenesis is crucial for developing effective and targeted therapies that address the underlying causes of inflammation and promote mucosal healing [90].

## 7. Molecular Mechanisms

### Cytokines and Chemokines

Macrophages play a pivotal role in immune responses within the GI tract by secreting a diverse array of cytokines and chemokines [52]. During the initial stages of infection and inflammation, these immune cells orchestrate complex signaling cascades that are essential for host defense and tissue repair. The key pro-inflammatory cytokines produced by macrophages are tumor necrosis factor-alpha (TNF-α), interleukin-1 beta (IL-1β), and interleukin-6 (IL-6). These cytokines serve as potent mediators that recruit and activate various immune cells to mount a robust defense against infection.

TNF-α enhances the inflammatory response by promoting leukocyte recruitment and activating endothelial cells to facilitate immune cell extravasation into affected tissues. This process ensures that a sufficient number of immune cells reach the site of infection, where they can effectively combat invading pathogens [91]. IL-1β acts synergistically with TNF-α to amplify the inflammatory response and induce fever, which is a critical component of the body’s defense mechanism against pathogens [92]. The fever response helps to inhibit microbial growth and enhances the efficiency of immune cell function. IL-6, on the other hand, plays a crucial role in the adaptive immune response by promoting B cell differentiation and antibody production. This enhances the body’s ability to recognize and neutralize microbial invaders, thereby providing a more targeted and effective immune response.

In addition to their pro-inflammatory functions, macrophages secrete anti-inflammatory cytokines such as IL-10 and TGF-β during the resolution phase of inflammation. These cytokines help suppress excessive inflammation, promote tissue repair, and restore homeostasis within the GI tract [93,94]. The ability of macrophages to dynamically regulate both pro-inflammatory and anti-inflammatory responses underscores their importance in maintaining immune balance and ensuring effective pathogen clearance, while preventing chronic inflammation and tissue damage.

## 8. Signaling Pathways

Central to macrophage activation are signaling pathways, such as the nuclear factor kappa-light-chain-enhancer of activated B cells (NF-κB) and mitogen-activated protein kinase (MAPK) [95]. These pathways govern diverse cellular processes, including cytokine production, antimicrobial peptide synthesis, and immune cell activation, thereby regulating the magnitude and duration of the inflammatory responses (Table 1). The NF-κB pathway plays a pivotal role in regulating the transcription of proinflammatory cytokines such as TNF-α and IL-1β in response to microbial stimuli [96]. The activation of NF-κB induces the expression of genes involved in immune and inflammatory responses, which are crucial for mounting an effective defense against pathogens.

Concurrently, the MAPK pathway modulates cellular responses to extracellular signals by integrating environmental cues to regulate macrophage functions. MAPK signaling cascades, including ERK, JNK, and p38 MAPK, mediate cellular processes, ranging from proliferation and differentiation to stress responses and immune activation [97]. In macrophages, the activation of MAPK pathways contributes to cytokine production and antimicrobial peptide synthesis, which are essential for coordinating immune responses and maintaining gut barrier integrity in response to microbial challenges.

The JAKSTAT pathway is also vital in macrophage activation, particularly in response to cytokines and growth factors [98]. This pathway transduces signals from the cell surface to the nucleus, leading to gene expression changes that influence macrophage survival, proliferation, and differentiation [99]. In the context of gut inflammation, the JAK/STAT pathway modulates the immune response and plays a crucial role in regulating inflammation and maintaining intestinal homeostasis.

The PI3K/Akt pathway regulates various cellular processes including metabolism, growth, and survival. In macrophages, the activation of the PI3K/Akt pathway enhances cell survival and promotes anti-inflammatory responses [100]. This pathway contributes to the resolution of inflammation by inhibiting pro-inflammatory cytokine production and promoting the expression of anti-inflammatory mediators.

Toll-like receptor (TLR) pathways are critical for innate immune responses. TLRs recognize pathogen-associated molecular patterns (PAMPs) and initiate signaling cascades that lead to the activation of the NF-κB and MAPK pathways. In macrophages, TLR signaling enhances the production of pro-inflammatory cytokines and antimicrobial peptides, facilitating a rapid response to microbial infections and maintaining gut barrier function [101].

**Table 1 ijms-25-09422-t001:** The molecular signaling pathways relevant to macrophage function and their implications for IBS.

Signaling Pathway	Components	Functions	Implications for IBS
NF-κB Pathway	-NF-κB (nuclear factor kappa-light-chain-enhancer of activated B cells)	-Regulates transcription of inflammatory cytokines.	-Central in regulating inflammatory responses to microbial stimuli.
	-TNF-α	-Induces expression of genes involved in immune and inflammatory responses.	-Crucial for mounting effective defenses against pathogens [102,103,104].
	-IL-1β		
MAPK Pathway	-ERK (extracellular signal-regulated kinase)	-Modulates cellular responses to extracellular signals.	-Coordinates immune responses.
	-JNK (c-Jun N-terminal kinase)	-Regulates processes like proliferation, differentiation, stress responses, and immune activation.	-Maintains gut barrier integrity in response to microbial challenges [105].
	-p38 MAPK	-Contributes to cytokine production and antimicrobial peptide synthesis.	
JAK/STAT Pathway	-JAK (Janus kinases)	-Mediates cytokine signaling.	-Influences macrophage activation and polarization.
	-STAT (Signal transducers and activators of transcription)	-Regulates gene expression related to immune responses and inflammation.	-Impacts inflammation and immune regulation in the gut [98,99].
PI3K/Akt Pathway	-PI3K (Phosphoinositide 3-kinases)	-Controls cell survival, growth, and metabolism.	-Modulates macrophage function and survival.
	-Akt (Protein kinase B)	-Regulates inflammatory responses and cytokine production.	-Affects inflammation and tissue repair processes in the gut [100].
Toll-Like Receptor (TLR) Pathway	-TLRs (Toll-like receptors)	-Recognizes pathogen-associated molecular patterns (PAMPs).	-Critical for recognizing microbial infections.
	-MyD88 (Myeloid differentiation primary response 88)	-Activates downstream signaling leading to cytokine production and inflammation.	-Drives inflammation and immune responses in the gut [101].
	-TRIF (TIR-domain-containing adapter-inducing interferon-β)		

## 9. MicroRNA Regulation

MicroRNAs (miRNAs) are critical regulators of gene expression in macrophages, influencing essential processes, such as cytokine production, inflammation resolution, and epithelial repair within the gastrointestinal (GI) tract (Table 2). The dysregulation of miRNA-mediated pathways in macrophages can significantly impact immune responses, contributing to the pathogenesis of various GI disorders, including irritable bowel syndrome (IBS) and inflammatory bowel disease (IBD).

One well-studied miRNA in this context is miR-29, which downregulates CLDN1 and NKRF, leading to increased intestinal permeability. This mechanism has been observed in knockout mice and intestinal tissue samples from patients with diarrhea-predominant IBS (IBS-D) [106]. Additionally, miR-146a and miR-155 have emerged as critical regulators of macrophage polarization and function. MiR-146a acts as a negative regulator of NF-κB signaling by targeting IRAK1 and TRAF6, thereby attenuating pro-inflammatory responses and promoting immune homeostasis [107]. In contrast, miR-155 enhances macrophage activation by targeting SOCS1 and SHIP1, amplifying inflammatory responses, and promoting pathogen clearance [108]. The dysregulation of miRNA-mediated pathways in macrophages can significantly affect immune responses and overall gut health, underscoring their potential as therapeutic targets in GI diseases. In patients with PI-IBS-D, colonic miR-155 and TNF-α levels were elevated, along with decreased colonic COMT expression (Figure 1). In COMT^−/−^ mice, miR-155 and TNF-α expression was significantly increased in both colon tissues and dorsal root ganglia. The administration of the cV1q antibody (anti-TNF-α) in these mice reversed visceral hypersensitivity induced by trinitrobenzene sulfonic acid (Figure 2) [86].

Recent studies have highlighted the roles of specific miRNAs in IBD, such as miR-331-3p and hsa-let-7d-5p [109]. miR-331-3p suppresses the expression of IL-12/IL-23p40 in macrophages, thereby reducing the production of these cytokines, which are key mediators of inflammation in IBD [110]. Thus, this miRNA plays a protective role by limiting excessive inflammatory responses in the gut. In contrast, hsa-let-7d-5p has been implicated in the regulation of immune responses in IBD by targeting various signaling pathways, including those involved in cytokine production and macrophage activation [111]. Alterations in the expression of hsa-let-7d-5p have been associated with the severity of inflammation in patients with IBD, making it a potential biomarker and therapeutic target [112].

In addition, miR-199 has been found to influence visceral pain in IBS patients by upregulating the translation of TRPV1, a receptor involved in pain perception [113]. The dysregulation of miRNAs, such as miR-21 and miR-124, further underscores the complexity of miRNA involvement in gut inflammation. MiR-21 modulates macrophage polarization by targeting PTEN, leading to enhanced Akt signaling and promoting an anti-inflammatory M2 phenotype [114]. MiR-124, on the other hand, suppresses macrophage activation by targeting STAT3 and CEBPα, reducing pro-inflammatory cytokine production, and promoting inflammation resolution [115,116].

Moreover, miR-223 plays a crucial role in controlling macrophage-mediated inflammation by negatively regulating the NLRP3 inflammasome activity. By targeting NLRP3 and IL-1β, miR-223 attenuates excessive inflammatory responses, thereby protecting against gut inflammation and associated tissue damage [117]. The role of miRNAs in modulating macrophage functions extends beyond individual miRNAs, as interactions between different miRNAs and their targets create a complex regulatory network that influences the overall inflammatory response and tissue repair processes.

Given the pivotal roles of miRNAs, such as miR-331-3p and hsa-let-7d-5p, in regulating macrophage activity and their involvement in IBD, harnessing these regulatory molecules offers promising avenues for developing targeted therapies. Such therapies can modulate macrophage function, promote tissue repair, and maintain intestinal homeostasis, ultimately providing new strategies for managing IBD and other inflammatory gut disorders.

**Table 2 ijms-25-09422-t002:** Summarizing the key miRNAs and their functions in IBS and IBD.

miRNA	Target Genes	Function	Implications for IBS and IBD
miR-29	CLDN1, NKRF	Downregulates target genes, increasing intestinal permeability.	Increased intestinal permeability in IBS-D patients [106].
miR-146a	IRAK1, TRAF6	Negative regulator of NF-κB signaling, attenuating pro-inflammatory responses, and promoting immune homeostasis.	Reduces inflammation and promotes immune balance [107].
miR-155	SOCS1, SHIP1	Enhances macrophage activation, amplifies inflammatory responses, and promotes pathogen clearance.	Elevated levels linked to increased inflammation in IBS-D patients [108].
miR-199	TRPV1	Upregulates TRPV1 translation when decreased, enhancing visceral pain.	Associated with increased visceral pain in IBS patients [113].
miR-21	PTEN	Modulates macrophage polarization by targeting PTEN, leading to enhanced Akt signaling.	Promoting an anti-inflammatory M2 phenotype [114].
miR-331-3p	IL-12/IL-23p40	Suppresses the expression of IL-12/IL-23p40 in macrophages, thereby reducing the production of these cytokines, which are key mediators of inflammation in IBD.	This miRNA plays a protective role by limiting excessive inflammatory responses in the gut [109,110].
let-7d-5p	LGALS3	LGALS3 silencing or LPS treatment blocked the TLR4/NF-κB signaling pathway.	Exercising its inhibitory properties in the inflammatory response via inactivation of the LGALS3-dependent TLR4/NF-κB signaling pathway [109,112,118].

### 9.1. Therapeutic Implications

Targeting macrophages presents a promising therapeutic strategy for managing GI diseases by modulating immune responses and promoting tissue repair. Small molecule inhibitors targeting the NF-κB or MAPK pathways can attenuate excessive inflammation and restore gut barrier integrity in conditions such as Crohn’s disease and ulcerative colitis [119]. Biologics, including monoclonal antibodies against pro-inflammatory cytokines such as TNF-α and IL-1β, offer targeted approaches to dampen immune responses and alleviate symptoms in patients with refractory IBS [86]. 

Furthermore, gene therapy approaches may exploit miRNA-mediated regulation to manipulate macrophage polarization and enhance therapeutic outcomes [86]. By targeting key miRNAs, such as miR-146a and miR-155, researchers can modulate inflammatory responses and promote tissue repair in a personalized manner, tailored to individual variations in genetic predisposition and microbiota composition. 

### 9.2. Microbiota-Based Therapies

Probiotics and prebiotics represent innovative microbiota-based therapies that influence macrophage function and improve gut health in GI disorders [120]. Probiotics, comprising live beneficial bacteria, interact with macrophages and other immune cells within the GI tract, promoting an anti-inflammatory environment through the production of short-chain fatty acids and the modulation of immune responses [121]. Prebiotics provide fermentable substrates for beneficial bacteria, enhancing their growth and metabolic activity, which in turn support gut homeostasis and immune function. These therapies offer dual benefits by modulating the composition and activity of the gut microbiota, while promoting macrophage-mediated immune responses and tissue repair. By restoring microbial balance and enhancing barrier function, probiotics and prebiotics represent promising strategies for managing inflammatory conditions and improving outcomes in patients with IBS and other GI diseases [122].

### 9.3. Current Challenges and Future Directions

Despite significant advancements, several challenges remain in understanding the complex interactions between macrophages, the gut microbiota, and GI health. Further research is needed to elucidate the intricate molecular mechanisms underlying macrophage responses to microbial stimuli and their roles in maintaining gut barrier integrity. This includes investigating the functional diversity of macrophage subtypes (e.g., M1 and M2) and their phenotypic plasticity in different physiological and pathological contexts within the GI tract [123]. Technological advances in omics technologies, including genomics, transcriptomics, proteomics, and metabolomics, offer unprecedented opportunities to comprehensively study macrophage–gut barrier interactions at the system level. Integrating omics data can reveal novel biomarkers associated with macrophage dysfunction in GI diseases, facilitating the development of targeted diagnostic tools and therapeutic strategies [124].

### 9.4. Precision Medicine Approaches

Personalized therapies based on individual variations in genetics, microbiota composition, and immune profiles are promising for optimizing treatment outcomes in GI disorders. Precision medicine approaches aim to tailor therapeutic interventions according to the unique biological characteristics of each patient, thereby maximizing efficacy and minimizing adverse effects [125]. By leveraging patient-specific data, clinicians can select optimal therapies that target macrophage-mediated pathways implicated in disease pathogenesis, paving the way for personalized management strategies in complex conditions, such as IBS [126]. These advancements underscore the evolving landscape of macrophage research in GI health and highlight the potential for transformative discoveries and personalized treatment options in the near future. By advancing our understanding of macrophage biology and leveraging cutting-edge technologies, we are poised to revolutionize the management of GI diseases, offering new hope for improved outcomes and quality of life in patients worldwide. These advancements and approaches underscore the evolving landscape of macrophage research in GI health and highlight the potential for transformative discoveries and personalized treatment options in the near future [127].

## 10. Conclusions

Macrophages play a pivotal role in maintaining gut barrier integrity and orchestrating essential post-inflammatory and post-infection responses within the GI tract (Figure 3). Their multifaceted functions, including the facilitation of epithelial repair, modulation of immune responses, and regulation of the microbial balance, are critical for GI health. However, although macrophage-targeted therapies offer promising avenues for managing gastrointestinal diseases, including complex conditions such as IBS, these approaches are not without significant challenges and risks.

One major challenge lies in the inherent plasticity of macrophages, which, while beneficial in response to various stimuli, also poses the risk of unintended consequences. For instance, the overactivation of the anti-inflammatory M2 phenotype could lead to fibrosis or impaired pathogen clearance, whereas the insufficient modulation of the pro-inflammatory M1 phenotype might result in chronic inflammation or tissue damage. Furthermore, the complex interactions among macrophages, gut microbiota, and the host immune system make it difficult to predict therapeutic outcomes and potential side effects.

Future research must prioritize the understanding of these complexities and identify strategies to precisely modulate macrophage activity without triggering adverse effects. Specific areas for further investigation include delineating the diverse functions of macrophage subtypes, understanding their interactions with the gut microbiota, and exploring how these interactions influence disease progression and treatment response. Leveraging cutting-edge omics technologies and advanced modeling techniques will be crucial for uncovering novel biomarkers and therapeutic targets as well as for developing safer and more effective interventions.

The potential for macrophage-targeted therapies to revolutionize the management of GI diseases is significant but requires a cautious and nuanced approach. The path to personalized therapeutic strategies, which tailor interventions to individual genetic, microbiota, and immune profiles, is promising but fraught with challenges that must be carefully addressed. Through concerted research efforts and technological advancements, we can work towards realizing the full potential of these therapies, while also ensuring patient safety and optimizing treatment outcomes.

## Figures and Tables

**Figure 1 ijms-25-09422-f001:**
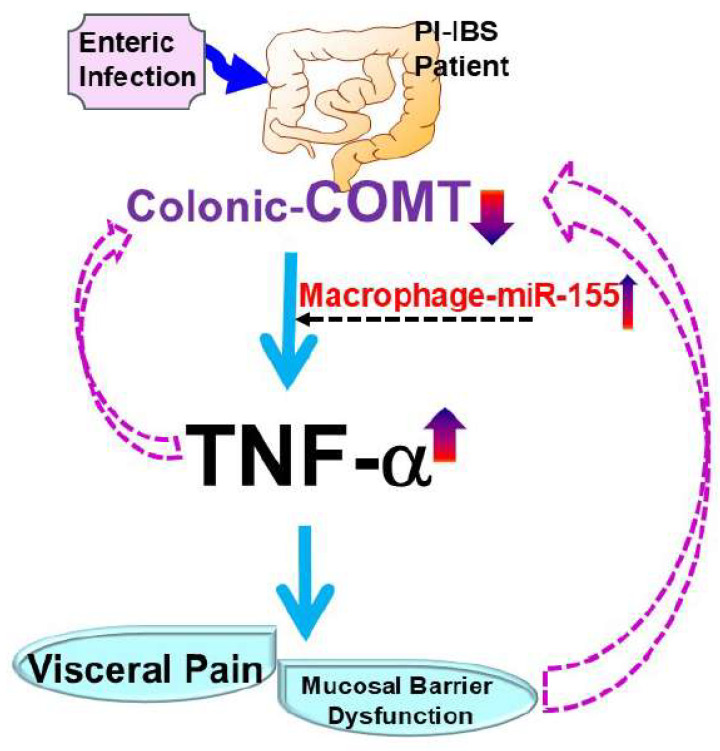
**Role of COMT–miRNA axis in macrophage-mediated mucosal barrier dysfunction in PI-IBS patients**. This figure elucidates the bioenvironment after an enteric infection; the dysregulation of COMT–miRNA-mediated pathways in macrophages can significantly affect immune responses and overall gut health, highlighting their potential as therapeutic targets in gastrointestinal diseases. In patients with PI-IBS-D, elevated levels of colonic miR-155 and TNF-α were observed, along with decreased colonic COMT expression.

**Figure 2 ijms-25-09422-f002:**
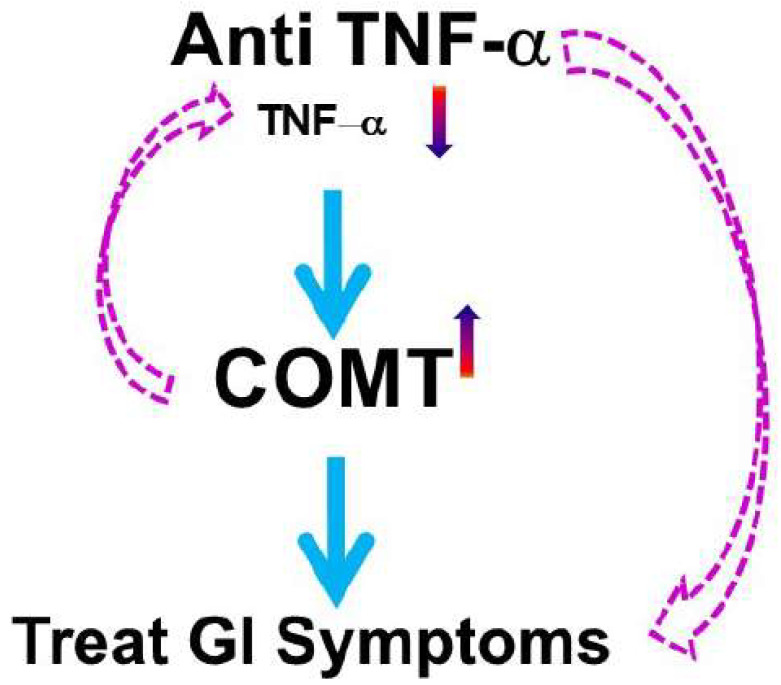
**Reversal of visceral hypersensitivity by anti-TNF−α therapy in an animal model**. This figure describes the mouse model that received anti-TNF−α and had a significant reduction in visceral sensitivity via feedback through an upregulated COMT mechanism.

**Figure 3 ijms-25-09422-f003:**
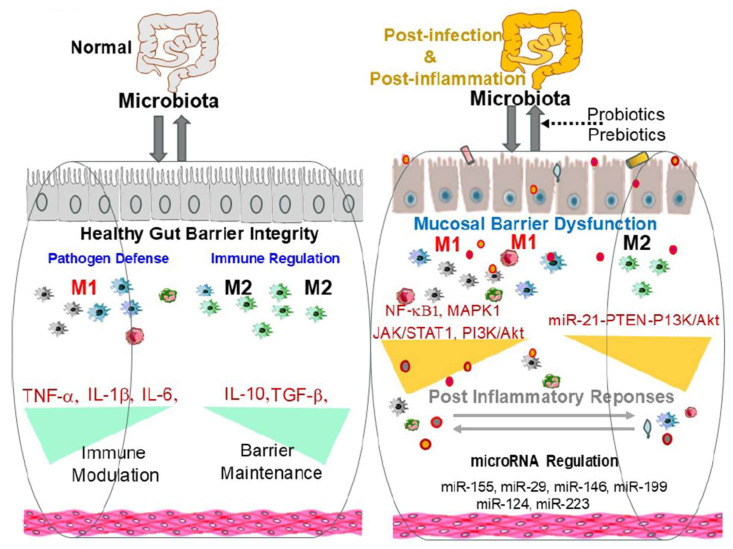
Summary cartoon: The pivotal role of macrophages in preserving gut barrier integrity and orchestrating vital responses following inflammation and infection within the gastrointestinal (GI) tract. Macrophages perform a variety of crucial functions that are essential for GI health. These include facilitating epithelial repair by secreting growth factors that promote cell proliferation and differentiation, modulating immune responses through the production of pro-inflammatory and anti-inflammatory cytokines, and regulating microbial balance by phagocytosing pathogens and producing antimicrobial peptides. Tissue-resident (M2) and infiltration macrophages (M1) switch the phenotypes in gut problems such as IBS via major signaling pathways mediated by NF-κB, JAK/STAT, PI3K/AKT, MAPK, Toll-like receptors, and specific microRNAs such as miR-155, miR-29, miR-146a, and miR-199. Together, these multifaceted roles ensure the maintenance of a healthy gut environment, effective pathogen clearance, and the restoration of homeostasis following GI disturbances.

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
