# Peer review of "Macrophages and Gut Barrier Function: Guardians of Gastrointestinal Health in Post-Inflammatory and Post-Infection Responses"

_ijms, 2024, doi:10.3390/ijms25179422_

Round 1

Reviewer 1 Report

Comments and Suggestions for Authors

The abstract of the paper effectively summarizes the multifaceted role of macrophages in maintaining gut barrier function and their involvement in post-inflammatory and post-infectious responses. However, it is somewhat verbose and could benefit from more concise language to improve clarity and focus. For instance, the repeated emphasis on macrophages' roles might be streamlined to provide a clearer, more direct summary of the paper’s objectives and conclusions.

The introduction should benefit from a clearer statement of the research gap or specific objectives, which are somewhat diluted in the current text.

The discussion of the interaction between the gut barrier and microbiota, while informative, could benefit from more recent references to enhance the relevance of the literature cited.

The section Role of Macrophages in Maintaining Gut Barrier Function is somewhat repetitive, particularly in describing the functions of resident and infiltrating macrophages. The authors might improve this section by focusing on the unique contributions of each macrophage type and reducing redundant descriptions

The authors should also consider discussing the potential side effects of modulating macrophage activity in Post-Inflammatory and Post-Infection Responses section, those side effects are not sufficiently addressed. This would provide a more critical perspective on the therapeutic potential discussed.

The conclusion tends to overstate the potential of macrophage-targeted therapies without sufficiently addressing the challenges and risks. Present a more balanced conclusion that acknowledges the challenges and suggests specific areas for future research. This would provide a more realistic and thoughtful wrap-up of the paper.

Overall the paper needs a more concise writing style, reducing redundancy and improving the logical flow of ideas.

Reviewer 2 Report

Comments and Suggestions for Authors

I read with interest this review on the role of macrophages in maintaining the integrity of the intestinal epithelial barrier.

I have some considerations for the authors:

1) I find sections 1 and 2 of this review need to be completely redone. This "shopping list" approach is very disorganized and confuses the reader. It would be better to create two paragraphs without lists in a more fluid manner and possibly accompany them with a figure if possible.

2) In the section on microRNAs, I would also include the role of miR-331-3p and hsa-let-7d-5p in IBD.

3) In the microbiota-based therapy section, I suggest including fungi as well as bacteria. I recommend the authors discuss emerging evidence regarding Hericium erinaceus, which, in ex vivo experiments, has shown the ability to downregulate pro-inflammatory cytokines that also influence macrophages (https://pubmed.ncbi.nlm.nih.gov/37465689/).

Reviewer 3 Report

Comments and Suggestions for Authors

The authors present a review of the involvement of macrophages in gut barrier function and their interaction with the gut microbiome with the example of the IBS disease state. 

This is an important area of research, where a comprehensive review of the topic would pose benefit to the research community. 

A number of vital improvements are needed to elevate the review from its current state. 

Broadly, the authors only mention IBS, why not other commonly encountered disease states? While important, only the one condition does not provide enough breadth for the applicability of the review's contents. 

Similarly, not enough information is provided regarding the interactions, both direct and indirect, between macrophages and the gut microbiome. This is indeed an important area, and reciprocal influence of each on the other would be of great benefit to the narrative of the review.

The layout of section 2 should be modified. While the content is fine, the division into numerous headings loses any logical flow to the information.

The style of writing has an introductory paragraph to each section and a conclusion. These should be removed as they are repetitive and do not add any further content. Just make the intended points. 

Also there is a great deal of repetition of points throughout - such as that of IL-10. Section 6.1 is suitable with its explanation. The repetitive sprinkling throughout should be minimised. The same can be said for pathogen clearance. 

Table 1 is under referenced.

Comments on the Quality of English Language

There are numerous typos throughout. Also ensure consistency in spacing and font size (section 4.2)

Round 2

Reviewer 2 Report

Comments and Suggestions for Authors

The authors have performed the relevant revisions. The manuscript can be accepted according to my opinion.

Author Response

We appreciate the reviewers comments that the paper is ready for publication.

Reviewer 3 Report

Comments and Suggestions for Authors

This revised version of the review has improved a lot. The flow is much better and the sections previously unclear are much easier to read. The additional information added has also improved the paper. 

My only criticism would be to remove the last paragraph of section 3.1 "In summary..." as it is not needed and seems out of place.

Author Response

We agree and have deleted the last paragraph of section 3.1 "In summary..." as it is not needed and seems out of place.